# Work Fatigue in a Hospital Setting: The Experience at Cheng Hsin General Hospital

**DOI:** 10.3390/healthcare9060776

**Published:** 2021-06-21

**Authors:** Tao-Hsin Tung, Ming-Chon Hsiung

**Affiliations:** 1Evidence-Based Medicine Center, Taizhou Hospital of Zhejiang Province Affiliated to Wenzhou Medical University, Linhai 317000, China; ch2876@yeah.net; 2Department of Occupational Safety and Health, Cheng-Hsin General Hospital, Taipei 112, Taiwan

**Keywords:** fatigue, medical staff, hospital setting

## Abstract

We aimed to investigate fatigue and its related factors in a medical professional population aged ≥30 years, as appraised by the implementation of an employee health screening program at Cheng Hsin General Hospital in Taipei, Taiwan. The study participants included a total of 2132 (400 males and 1732 females) healthy medical professionals enrolled in a teaching hospital who underwent physical verification in 2019. Demographic characteristics and fatigue information were collected. The overall prevalence of personal- and work-related fatigue in this study population was 41.4% and 39.1%, respectively. The prevalence of a high risk of work- or personal-related fatigue proved to be substantially greater (*p*-value for chi-square test <0.0001) than it was for a low or moderate risk of personal-related fatigue. Using multinominal logistic regression analysis, seniority and position were statistically significant in relation to a high risk of personal- and work-related fatigue. Personal- and work-related fatigue were found to be prevalent in physicians and nurses. Lower seniority was also related to severe personal- or work-related fatigue. Providing this population with controlled working environments and health improvements is important.

## 1. Introduction

A sustained period of demanding physical, intellectual, or emotional activity results in the perception of fatigue [1]. The definition of fatigue is a “work-related condition that ranges from acute to chronic in nature and can produce an overwhelming sense of tiredness, decreased energy, and exhaustion resulting in impaired physical functioning, cognitive functioning, or both” [2]. Physical and mental health are both significantly related to an individual’s biological, psychological, and cognitive progression [3]. Evidence-based studies have indicated that fatigue frequently occurs in workplace settings, with 23–40% of workers revealing high levels of fatigue [4]. Among the general working population, more than 60% feel somewhere between slightly tired to completely exhausted after a working day [5]. Healthcare professionals are at high risk of exposure to posture-related injury, which can cause disorders of the musculature and skeleton and can become a primary factor in physical fatigue. Thus, these high rates of fatigue among clinical professionals must be given attention as a significant problem.

Healthcare professionals often spend much time standing and performing activities requiring physical effort; they may experience more fatigue due to the direct effects of physical activity [6]. Fatigue can also decrease work attendance and may undermine workers’ ability to behave properly in the workplace [7]. From the viewpoint of preventative medicine, it is not only necessary to be aware of the risk of fatigue in itself but also to examine the full extent of factors that may be associated with high fatigue levels. Furthermore, some uncertainty remains regarding the factors that correlate with elevated fatigue, which show professional disparity among a sub-population in Taiwan. Thus, to determine the current situation and to identify the factors related to high levels of fatigue, this study aimed to identify the factors related to fatigue to further our knowledge of fatigue. This study also aimed to investigate the prevalence of fatigue and its related factors amongst a population of medical professionals aged ≥30 years. This was achieved through the implementation of an employee health screening regime at Cheng Hsin General Hospital, a fully approved regional teaching hospital in Taipei, Taiwan.

## 2. Methods

### 2.1. Study Design and Data Selection

In 2019, 2612 healthcare professionals were employed at Cheng Hsin General Hospital, an officially recognized regional teaching hospital with 1030 beds in Taipei, Taiwan. According to Taiwan’s labor and health regulations, this cross-sectional study examined a total of 2132 medical staff (400 males and 1732 females) admitted to Cheng Hsin General Hospital for a routine health examination between January 2019 and December 2019. The coverage rate was 81.6%. All courses were applied in keeping with the principles of our institutional ethics committee and in accordance with the Declaration of Helsinki. All of the participants’ data were kept anonymous. This study was approved by the Institution Review Board of Cheng Hsin General Hospital on 6 February 2021 (CHGH-IRB No: (805)109A-44).

### 2.2. Measures

This study used closed-ended question sheets, which were created based on the qualitative access outcomes and associated studies and research projects. The details of the questionnaire are as follows:

Demographic characteristics, which included sociodemographic variables (gender, age, seniority, and professional position) and health management (normal or abnormal status which screened by physicians).

Personal-related fatigue, which consists of six items [8]. A five-point Likert scale was used (0 means never and 100 means always). Then, we added up the scores of questions 1–6 and divided it by 6 to obtain a personal-related fatigue score. A value of < 50, 50–70, or ≥ 70 was considered to represent low, moderate, or high risk of personal-related fatigue. In the present study, the results indicated good internal reliability (Cronbach’s α = 0.84) of this scale.

Work-related fatigue, which consists of seven items [8]. A five-point Likert scale was applied (0 means never and 100 means always). The seventh item was a reverse question. Then, we added up the scores of questions 1–7 and divided them by 7 to obtain a work-related fatigue score. A value of < 45, 45–60, or ≥ 60 was considered to represent low, moderate, and high risk of work-related fatigue. In the present study, the results indicated good internal reliability (Cronbach’s α = 0.87) of this scale.

### 2.3. Data Analysis

Statistical analysis was carried out using SAS for Windows (SAS version 9.1; SAS Institute Inc., Cary, NC, USA). In an unadjusted analysis, the chi-square (*χ*^2^) test was performed to explore categorical variables between subjects with and without fatigue. Multinomial logistic regression was further used for the logistic regression with binary outcomes when the categorical dependent variable was more than two levels [8]. This approach was also applied to indicate a series of statistical coefficients for each of the two comparisons of fatigue and assess the independent factors related to the prevalent fatigue. A *p*-value of < 0.05 was considered to show a statistically significant difference among the study samples.

## 3. Results

The overall prevalence of personal-related fatigue (moderate risk, 28.2%; high risk, 12.5%) and work-related fatigue (moderate risk: 24.9%; high risk: 14.2%) for the study participants was 41.4% and 39.1%, respectively. Figure 1 shows that the prevalence of a high risk of work- or personal-related fatigue proved to be substantially greater (*p*-value for *χ*^2^ test < 0.001) than it was for a low or moderate risk of personal-related fatigue.

Table 1 shows the findings for various test variables and their potential relationship with personal- or work-related fatigue (yes or no) for the study subjects. Based on the *χ*^2^ test, the related factors significantly associated with personal-related fatigue included female gender, young age, being a physician or a nurse, and abnormal health management. Meanwhile, factors including female gender, young age, low seniority, and being a physician or a nurse were associated with factors related to work-related fatigue.

The effects of independent related factors on different types of personal- and work-related fatigue were examined by using the multinominal logistic regression model. As indicated in Table 2, for personal-related fatigue, controlled for confounding factors, position (physician (OR = 2.47, 95%CI: 1.27–4.79), nurse (OR = 3.07, 95%CI: 1.81–5.20), nurse practitioner (OR = 2.76, 95%CI: 1.50–5.08), administration (OR = 1.89, 95%CI: 1.12–3.20), and health management (normal vs. abnormal, OR = 0.81, 95%CI: 066–0.99) appeared to be statistically significantly related to a moderate risk of personal-related fatigue. Seniority (>20 vs. <5 years, OR = 0.61, 95%CI: 0.39–0.96) and professional position (physician (OR = 3.33, 95%CI: 1.26–8.00), nurse (OR = 4.48, 95%CI: 1.98–10.15), or a nurse practitioner (OR = 2.92, 95%CI: 1.16–7.37)) were revealed to be statistically significantly associated with a high risk of personal-related fatigue.

In addition, Table 2 also shows that position (physician (OR = 3.35, 95%CI: 1.67–6.68), nurse (OR = 3.75, 95%CI: 2.11–6.64), or nurse practitioner (OR = 2.78, 95%CI: 1.44–5.35) appeared to be statistically significantly related to a moderate risk of work-related fatigue. Seniority (>20 vs. <5 years, OR = 0.53, 95%CI: 0.34–0.81; 15–20 vs. <5 years, OR = 0.59, 95%CI: 0.38–0.93) and position (physician (OR = 3.30, 95%CI: 1.17–9.31), nurse (OR = 7.44, 95%CI: 3.12–17.70), nurse practitioner (OR = 4.09, 95%CI: 1.56–10.73), or medical personnel (OR = 2.59, 95%CI: 1.04–6.45)) appeared to be statistically significantly related to a high risk of work-related fatigue.

## 4. Discussion

Fatigue involves feelings of overtiredness, a lack of vigor, and prostration. This situation is not only frequently experienced by workers occupied in daily work but also affects both physical and cognitive functioning [5]. Previous studies have indicated that fatigue is more serious in the medical profession than in the general population. The particular occupational groups of nurses and physicians have more elevated degrees of fatigue than health service directors and administrative personnel [6,9]. It is estimated that the health-related and economic effects of fatigue in working populations are tremendous and that fatigued workers, cost employers $101 billion annually more than non-fatigued workers in health-related lost working time [4,10]. If fatigue is not relieved, it may further lead to harmful consequences, such as lower work engagement, higher sickness absence, and intention to quit [11,12].

Work-related fatigue is caused by physiological, cognitive, emotional, and sensory elements as a consequence of high work volume and insufficient time allowed for energy recovery [13]. Not surprisingly, this study indicated that there was a significant relationship between seniority and a high risk of work-related fatigue. Previous results have shown that, when fatigued, new nurses develop suboptimal intentions toward their clinical work, which can be detrimental to their professional and psychological activity over time [11]. Older medical staff with higher medical/technical skills often serve as “team leaders” and make decisions concerning patients, whereas younger staff carry out these decisions via physical labor [14]. To reduce work-related fatigue, it is necessary to explore the nature and demands of the work performed within each class of seniority. It could be worth offering a continuous education training program to cultivate independence and provide substantive motivation for low-seniority faculty.

Consistent with previous results [6,15,16,17], our study revealed that nurses or physicians are more likely to experience personal- or work-related fatigue. This suggests that professional position is an indicator for the deterioration of fatigue after adjustment for confounding factors. Evidence-based studies have indicated that work requiring 24 h coverage more easily reduces performance and increases occupational accidents and sickness among employees, especially under conditions where timetables are made in ways that violate human circadian rhythms [18]. Physicians and nurses who work during the night risk an increased probability of patient care errors than those who work during the day [19]. Working longer than 40 h per week, including voluntarily paid overtime, is associated with unfavorable incidents, such as medication errors, patient falls, and hospital infections [20,21].

It may be valuable to re-arrange tasks where feasible so that clinical professionals have more command or perform more recompensing work later in the working day, which could serve to improve motivation, decrease fatigue, improve patient safety, and reduce adverse medical events [6]. Clinical facility managers should create better working environments to improve fatigue based on the findings of this study. Managers must pay attention to the specific behaviors of employees and the requirements of various departments, establish goals and expectations for employees, increase their support for employees, and appropriately allocate organizational resources.

Satisfactory and trustworthy job environments must be established for medical staff to feel supported and encouraged in their professional duties. Primary supervisors should provide prompt personal care and intellectual encouragement, improve the job environment, and adjust their work to provide them with a reasonable workload. Work content (i.e., shift lengths and timing and task arrangement) must be adjusted to satisfy procedural justice. In addition, to adjust the workload, process reengineering should also be considered. As the characteristics of each department vary, the design and improvement of job procedures should be reevaluated from the perspectives of various departments. Medical jobs are professional service occupations. The scope of services is such that some services emphasize technical characteristics. For example, various technical operations have resulted in unified operational procedures at all levels. By contrast, some services, such as discharge care and shift handovers, accentuate service characteristics. Such service procedures should be re-engineered to reduce unnecessary costs and to improve job performance. As workplaces and job characteristics are oriented toward protecting the health of human beings, the personal, physical, and psychological health of medical staff should be the starting point for optimizing their job performance. Reducing the likelihood of medical staff sustaining diseases and experiencing discomfort should considerably lower their rate of absence and fatigue from carrying out their duties.

## 5. Limitations

Several methodological limitations should be mentioned when interpreting the findings of this study. First, in the selection of the study population—that is, healthcare professionals not only likely present a selection bias, but also the Hawthorne effect is certainly shown by the subjects who made the impartial decision to be in the study hospital. Voluntary bias originating from a specific sample could involve only those subjects who are actually ready to partake in the study and who join and find the subject especially enjoyable being more likely to volunteer for the study, identical to those who are prospectively assessed on an affirmative level [22]. Second, we designed the assessments at only a single time point, which is unable to indicate prolonged exposure to fatigue. Third, based on previous findings, income, marital status, and number of children display a link with fatigue [23]. However, it is difficult to control for these important variables due to the lack of information in the health examination database. The interpretation of results is thus limited in some respects. Fourth, the higher correlation between work- and personal-related fatigue level tells us that there is either personal bias, response bias, or that they overlap. 

Further studies should be conducted to explore possible intercorrelation—that is, that one is the determinant of the other. Finally, our study only included subjects from one teaching hospital in northern Taiwan as the target population. Therefore, the findings cannot be generalized to hospitals in other regions of Taiwan. Future epidemiological and follow-up investigations with larger study sample sizes of hospitals over a wider range of areas would make these results more convincing.

## 6. Conclusions

In conclusion, personal- and work-related fatigue were more prevalent among physicians and nurses. Lower seniority was also related to severe personal- or work-related fatigue. Therefore, helping this population through controlled working environments and health improvements is important.

## Figures and Tables

**Figure 1 healthcare-09-00776-f001:**
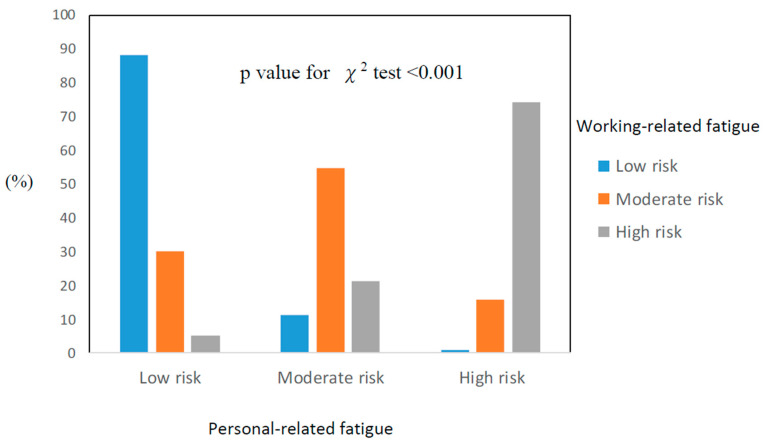
The association between personal-related fatigue and working-related fatigue among the study population (*n* = 2132).

**Table 1 healthcare-09-00776-t001:** The prevalence of each type of fatigue among the study population (*n* = 2132).

Variables	Types of Fatigue						
	Personal-Related				Working-Related		
	Total	Prevalence		*p*-Value for	Prevalence		*p*-Value for
	Number	Number	(%)	χ^2^-Test	Number	(%)	χ^2^-Test
**Gender**							
Male	400	128	(32.0)	<0.001	121	(30.2)	<0.001
Female	1732	754	(43.5)		712	(41.1)	
**Age (yrs)**							
<40	1051	495	(47.1)	<0.001	487	(46.3)	<0.001
40–49	621	259	(41.7)		233	(37.5)	
50–59	294	90	(30.6)		81	(27.6)	
≥60	166	38	(22.9)		32	(19.3)	
**Seniority (yrs)**							
<5	773	324	(41.9)	0.23	328	(42.4)	0.02
5–10	441	192	(43.5)		179	(40.6)	
10–15	314	134	(42.7)		122	(38.9)	
15–20	268	112	(41.8)		93	(34.7)	
≥20	336	120	(35.7)		111	(33.0)	
**Position**							
Supervisor	79	22	(27.8)	<0.001	19	(24.1)	<0.001
Physician	88	43	(43.4)		42	(42.4)	
Nurse	829	430	(51.9)		435	(52.5)	
Nurse practitioner	160	76	(47.5)		68	(42.5)	
Medical personnel	283	87	(30.7)		86	(30.4)	
Administration	636	219	(34.4)		180	(28.3)	
Part-time position	46	5	(10.9)		3	(6.5)	
**Health management**							
Normal	1315	521	(39.6)	0.04	499	(37.9)	0.18
Abnormal	817	361	(44.2)		334	(40.9)	
**Total**	2132	882	(41.4)		833	(39.1)	

**Table 2 healthcare-09-00776-t002:** Multinominal logistic regression of associated factors for personal-related and working-related fatigue that all univariate significant factors were included among screened subjects (*n* = 2132).

	Personal-Related						Working-Related					
	Moderate Risk vs.			High Risk vs.			Moderate Risk vs.			High Risk vs.		
Factors	Low Risk			Low Risk			Low Risk			Low Risk		
	**OR**	**95%CI**	***p*-Value**	**OR**	**95%CI**	***p*-Value**	**OR**	**95%CI**	***p*-Value**	**OR**	**95%CI**	***p*-Value**
**Gender**												
female vs. male	1.19	0.89–1.58	0.25	1.40	0.90–2.18	0.14	1.21	0.88–1.65	0.24	1.03	0.69–1.53	0.91
**Seniority (yrs)**												
>20 vs. <5	0.89	0.66–1.20	0.44	0.61	0.39–0.96	0.03	0.84	0.61–1.15	0.28	0.53	0.34–0.81	0.003
15–20 vs. <5	1.11	0.81–1.53	0.52	0.92	0.59–1.45	0.73	0.89	0.64–1.25	0.52	0.59	0.38–0.93	0.02
10–15 vs. <5	1.11	0.82–1.50	0.52	1.03	0.69–1.56	0.87	1.04	0.75–1.42	0.83	0.77	0.51–1.15	0.20
5–10 vs. <5	1.12	0.85–1.46	0.43	1.04	0.73–1.49	0.83	0.98	0.74–1.31	0.91	0.92	0.66–1.29	0.64
**Position**												
Supervisor or Part-time	1.00	-	-	1.00	-	-	1.00	-	-	1.00	-	-
Physician	2.47	1.27–4.79	0.007	3.33	1.26–8.80	0.02	3.35	1.67–6.68	0.001	3.30	1.17–9.31	0.02
Nurse	3.07	1.81–5.20	<0.001	4.48	1.98–10.15	<0.001	3.75	2.11–6.64	<0.001	7.44	3.12–17.70	<0.001
Nurse practitioner	2.76	1.50–5.08	0.001	2.92	1.16–7.37	0.02	2.78	1.44–5.35	0.002	4.09	1.56–10.73	0.004
Medical personnel	1.39	0.79–2.46	0.26	1.65	0.69–3.97	0.26	1.62	0.88–3.00	0.12	2.59	1.04–6.45	0.04
Administration	1.89	1.12–3.20	0.02	1.21	0.52–2.83	0.66	1.67	0.94–2.96	0.08	1.89	0.78–4.59	0.16
**Health management**												
Normal vs. Abnormal	0.81	0.66-0.99	0.04	0.81	0.61–1.07	0.13	0.87	0.71–1.08	0.21	0.77	0.59–1.01	0.06

## Data Availability

The raw data supporting the conclusions of this article will be made available by the authors, without undue reservation.

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
