# Peer review of "Work Fatigue in a Hospital Setting: The Experience at Cheng Hsin General Hospital"

_healthcare, 2021, doi:10.3390/healthcare9060776_

Round 1

Reviewer 1 Report

This study identifies that certain healthcare occupation groups, nurse and physicians have both types of fatigue, at personal level and at work-related level. However, the articles scientific contribution is not that significant. I have the following comments.

1) Why introduce with shift work when there is no control or statistical analysis of the same. Is it that shift work determines both type of fatigues? The class of workers with high fatigue are more into shift work?  

2) Important variables like income, marital status, number of children are not controlled for.

3) The higher correlation between work-related and personal-related fatigue level tells that there is either personal bias, response bias, or they overlap (which implies one is the determinant of the other and needs to be controlled for).

4) The findings have little significance or contribution. The contribution needs to be underlined. t can be so well, that certain healthcare workers are fatigued more than others. And this goes for both sources of fatigue. That means work has no extra role or the other way.

5) English needs to be edited in few places, though that is minor. Similarly the table needs to be formatted as headings and the columns are not aligned properly.

Reviewer 2 Report

Overall the paper is excellently written, it is concise and to the point. Fatigue in workplace is a very topical area for research and therefore this paper presents very insightful findings that add to the body of knowledge in the area. One suggestion is for the definition of fatigue to come early in the introduction rather than later in the discussion section.

Reviewer 3 Report

Dear authors,

Your manuscript is interesting but I need you to answer some questions:

INTRODUCTION

  • The introduction is very short. The constructs and concepts necessary to understand the manuscript are not explained.

METHODS

Study design and data selection:

  • What was the target population? How was the sample chosen? The authors must specify it.
  • The authors must include the response rate of the participants in the study.
  • The authors say: "This cross-sectional study was conducted with a total of 2132 medical staff (400 males and 1732 females) voluntarily admitted to Cheng Hsin General Hospital".

DISCUSSION

  • The Discussion must not have subsections. There must be a coherent and logical discourse in line with the Results.

REFERENCES

  • Many bibliographies are obsolete. The bibliographic citations used are more than 5 years old (58.8 %). The authors must update and arrange the bibliography.
  • There is an updated bibliography of original and meta-analytic articles that should be cited, among others.
  • Some references do not meet the journal guidelines.

Round 2

Reviewer 3 Report

Dear authors,

Thanks for your reply. The explanations of the authors are satisfactory. The paper has greatly improved its quality.

Congratulations on your work.

Best regards